# *In vitro* activities of contezolid (MRX-I) against drug-sensitive and drug-resistant *Mycobacterium tuberculosis*

Huiru An,[1] Wenna Sun,[1] Xiao Liu,[1] Tianhao Wang,[2] Juan Qiao,[3] Jianqin Liang[1]

**ABSTRACT**   A novel oxazolidinone for the treatment of *Mycobacterium tuberculosis* has been developed, but the activity of contezolid (MRX-I) still needs to be clarified. In this study, we isolated *Mycobacterium tuberculosis* from 48 clinical patients with pulmonary tuberculosis. Roche drug susceptibility tests identified drug-sensitive and 39 drug-resistant *M. tuberculosis* isolates. Drug susceptibility assays indicated that MRX-I exhibited anti-*Mycobacterium tuberculosis* activity against both drug-sensitive and drug-resistant isolates, with an advantage against drug-resistant isolates. The results also showed that the anti-*Mycobacterium tuberculosis* activity was comparable to that of linezolid.

**IMPORTANCE** Currently, *Mycobacterium tuberculosis* has exhibited increased drug resistance, leading to ineffective drug treatment in many patients with tuberculosis. Among the anti-*Mycobacterium tuberculosis* drugs, oxazolidinones have been gradually developed. Contezolid (MRX-I) has been newly developed in China with advantages versus the first oxazolidinone antibiotic approved by the Food and Drug Administration for clinical use, but the anti-*M. tuberculosis* activity needs to be further clarified. In this study, in vitro activities of MRX-I against *M. tuberculosis* were tested. The drug susceptibility assays indicated that MRX-I exhibited anti-*M. tuberculosis* activity comparable to that of linezolid, with an advantage against drug-resistant isolates.

**KEYWORDS**   *Mycobacterium tuberculosis*, contezolid, *in vitro*, drug resistant

The widespread use of various antibiotics has inevitably led to the increasingly serious problem of bacterial drug resistance, and the emergence of multidrug-resistant bacteria poses a major challenge to the public health systems worldwide. *Mycobacterium tuberculosis*, the pathogenic agent of pulmonary tuberculosis (TB), has exhibited increased drug resistance, leading to ineffective drug treatment in many patients with tuberculosis (1, 2). Among the anti-*Mycobacterium tuberculosis* drugs, oxazolidinones, a novel synthetic antibacterial drug class, have been gradually developed (3–5). Among the oxazolidinones, linezolid (LZD) has a strong antibacterial effect and was the first oxazolidinone antibiotic approved by the Food and Drug Administration for clinical use (6–8). However, LZD use can cause adverse reactions, such as anti-tuberculosis drug-induced liver injury, bone marrow suppression, and peripheral neuropathy (9).

Contezolid (MRX-I), a new oxazolidinone independently developed in China, has been approved for distribution on the clinical market. According to National Medical Products Administration (NMPA), MRX-I was recommended for the treatment of acute skin and soft-tissue infections caused by Gram-positive bacteria, including methicillin-resistant *Staphylococcus aureus* and vancomycin-resistant *Enterococcus*. MRX-I had a strong anti-Gram-positive bacterial ability with a similar efficacy to that of LZD, and the clinical cure rates of MRX-I and LZD were reported to be 93.0% and 93.4% at post-treatment visits, respectively, in a critical phase III study (CTR20150855) for the treatment of complex skin and soft-tissue infections in China which was conducted in 50

Address correspondence to Tianhao Wang, wt19750728@hotmail.com, Juan Qiao, qiaojuan775008@sina.com, or Jianqin Liang, ljqbj309@163.com.

Huiru An, Wenna Sun, and Xiao Liu contributed equally to this article. Author order was determined by drawing straws.

The authors declare no conflict of interest.

See the funding table on p. 4.

**TABLE 1** MICs of MRX-I compared to those of INH, RFP, SM, EMB, LFX, MXF, CFZ, CS, and LZD against nine isolates of drug-sensitive *Mycobacterium tuberculosis*[a]

| *Mycobacterium tuberculosis* isolate | MIC (µg/mL) | | | | | | | | | |
|---|---|---|---|---|---|---|---|---|---|---|
| | INH | RFP | SM | EMB | LFX | MXF | CFZ | CS | LZD | MRX-I |
| 215,680 | 0.025 | 0.25 | 0.5 | 0.625 | 0.25 | 0.125 | 0.125 | 4 | 0.25 | 0.5 |
| 220,213 | 0.025 | 0.125 | 0.25 | 0.625 | 0.25 | 0.125 | 0.125 | 4 | 0.25 | 0.25 |
| 220,220 | 0.025 | 0.125 | 0.25 | 0.625 | 0.25 | 0.125 | 0.125 | 8 | 0.25 | 0.5 |
| 220,311 | 0.025 | 0.125 | 0.25 | 0.625 | 0.25 | 0.125 | 0.25 | 8 | 0.5 | 0.5 |
| 220,391 | 0.025 | 0.125 | 0.25 | 0.625 | 0.25 | 0.125 | 0.25 | 4 | 0.25 | 0.25 |
| 220,412 | 0.025 | 0.125 | 0.25 | 0.625 | 0.25 | 0.125 | 0.125 | 4 | 0.25 | 0.25 |
| 220,450 | 0.025 | 0.125 | 0.25 | 0.625 | 0.25 | 0.125 | 0.125 | 8 | 0.5 | 0.5 |
| 220,509 | 0.025 | 0.25 | 0.25 | 0.625 | 0.25 | 0.125 | 0.125 | 2 | 0.5 | 0.5 |
| 220,667 | 0.2 | 1 | 0.5 | 0.625 | 0.25 | 0.125 | 0.25 | 8 | 0.25 | 0.5 |

[a]MRX-I, conezolid; INH, isoniazid; RFP, rifampicin; SM, streptomycin; EMB, ethambutol; LFX, levofloxacin; MXF, moxifloxacin; CFZ, clofazimine; CS, cycloserine; LZD, linezolid.

clinical centers (10, 11). Besides, compared with LZD, MRX-I has more safety advantages, including tolerance and minimal adverse effects (such as hematological adverse events and myelosuppression) (12). Additionally, it has been reported that MRX-I exhibits anti-*M. tuberculosis* activity *in vitro/vivo* and rarely induces drug resistance, so it could be a prospective drug for *M. tuberculosis* infection treatment (13). However, for drug-resistant *M. tuberculosis*, the efficacy of and susceptibility to MRX-I remain to be clarified.

This study was approved by the Ethics Committee of the Eighth Medical Center of PLA General Hospital (approval number: C2022-001). Informed consent was obtained from each patient. Clinical *M. tuberculosis* isolates were collected from 48 different patients with pulmonary tuberculosis. The specimen types included lung lavage fluid and sputum. Sample collection, pre-treatment, inoculation, identification, and drug susceptibility test were all in accordance with the "Laboratory Testing Procedures for Tuberculosis Diagnosis" assigned by Chinese Antituberculosis Association of Basic Medicine (14).

The samples were treated by alkaline treatment and neutralization centrifugal precipitation. The modified Roche medium used for culture was composed of 7.2 g sodium glutamate, 2.4 g $KH_2PO_4$, 2.4 g $MgSO_4 \cdot 7H_2O$, 0.6 g magnesium citrate, 12 mL glycerol, 600 mL ddH$_2$O, 1,000 mL fresh egg liquid, and 20 mL malachite green (2%).

Then the tested drugs were added into the improved Roche culture medium. A volume of 0.01 mL of each sample was taken and evenly inoculated onto the inclined surface of each culture tube for 24 h. Afterward, they were cultured vertically at 37°C for 4–6 weeks. The drug-sensitive *M. tuberculosis* isolates were then subjected to minimum inhibitory concentration (MIC) assays using the absolute concentration method. LZD and MRX-I showed consistent data for $MIC_{50}$ and $MIC_{90}$ of *M. tuberculosis* isolates, with values of 0.25 and 0.5 ug/mL, respectively. All the MICs values showed in this study were $MIC_{90}$. Apart from LZD and MRX-I, the drug susceptibility tests included four common first-line anti-*M. tuberculosis* drugs, isoniazid (INH), streptomycin (SM), ethambutol (EMB), and rifampicin (RFP) and eight other second-generation anti-*M. tuberculosis* drugs, including levofloxacin (LFX), kanamycin (KAN), capreomycin (CPM), prothionamide (TH1321), p-aminosalicylic acid (PAS), rifapentine (RFT), rifabutin (RBU), and amikacin (AMK).

It was found that of the 39 drug-resistant *M. tuberculosis* isolates, which were resistant to one or more of the agents tested, 82.0% (32/39) exhibited drug resistance to first-line anti-*M. tuberculosis* drugs, and 100.0% (39/39) exhibited drug resistance to the other second-generation drugs.

According to the Roche drug susceptibility test results of 48 clinical isolates, 9 isolates were pan-drug sensitive. The MIC of MRX-I ranged from 0.25 to 0.5 µg/mL. In addition, compared with another oxazolidinone, LZD and MRX-I exhibited anti-*M. tuberculosis* activity that was comparable to that of LZD. The MRX-I MIC value was higher than those of the other drugs, except cycloserine (CS) and EMB, for the drug-sensitive *M. tuberculosis* isolates. This result indicated that MRX-I had no significant advantage in the treatment of drug-sensitive *M. tuberculosis* (Table 1).

**TABLE 2** MICs of MRX-I compared to those of INH, RFP, SM, EMB, LFX, MXF, CFZ, CS, and LZD against 39 isolates of drug-resistant *Mycobacterium tuberculosis* isolate[a]

| | MIC (µg/mL) | | | | | | | | | |
|---|---|---|---|---|---|---|---|---|---|---|
| | INH | RFP | SM | EMB | LFX | MXF | CFZ | CS | LZD | MRX-I |
| 191,160 | 0.05 | 0.25 | 0.5 | 1.25 | 0.25 | 0.125 | 0.125 | 4 | 0.125 | 0.5 |
| 191,163 | 1.6 | 0.5 | 0.5 | 0.63 | 0.25 | 0.125 | 0.125 | 4 | 0.125 | 0.5 |
| 191,186 | 0.8 | >16 | >32 | 0.63 | 0.125 | 0.125 | 0.125 | 2 | 0.125 | 0.25 |
| 191,191 | 0.025 | 0.125 | 0.25 | 0.63 | 0.25 | 0.125 | 0.125 | 4 | 0.125 | 0.25 |
| 191,209 | 0.025 | 0.25 | 0.25 | 0.63 | 0.25 | 0.125 | 0.125 | 4 | 0.125 | 0.25 |
| 191,247 | 1.6 | 0.5 | 0.5 | 2.5 | 0.125 | 0.125 | 0.125 | 4 | 0.125 | 0.5 |
| 191,269 | 0.025 | 0.125 | >32 | 0.63 | 4 | 2 | 0.125 | 4 | 0.125 | 0.25 |
| 191,396 | 1.6 | >16 | >32 | 0.63 | 0.125 | 0.125 | 0.125 | 4 | 0.125 | 0.25 |
| 191,524 | 3.2 | >16 | 32 | 2.5 | 0.25 | 0.125 | 0.125 | 4 | 0.125 | 0.25 |
| 191,554 | 0.025 | 0.125 | 0.25 | 0.63 | 0.5 | 0.25 | 0.125 | 4 | 0.125 | 0.25 |
| 201,466 | >3.2 | 2 | >32 | 5 | 2 | 1 | 0.5 | 4 | 0.5 | 0.125 |
| 213,829 | 3.2 | 0.125 | 8 | 0.63 | 2 | 1 | 0.125 | 4 | 0.125 | 0.5 |
| 214,014 | 3.2 | 0.25 | >32 | 10 | 1 | 0.125 | 0.5 | 4 | 0.5 | 0.125 |
| 214,022 | 0.8 | 8 | 2 | 5 | 2 | 1 | 0.125 | 2 | 0.125 | 0.125 |
| 214,124 | 0.8 | >16 | >32 | 5 | 4 | 2 | 0.125 | 4 | 0.125 | 0.125 |
| 214,225 | 0.1 | 16 | >32 | 5 | 2 | 0.5 | 0.125 | 4 | 0.125 | 0.5 |
| 214,320 | 0.025 | >16 | 0.25 | 5 | 2 | 0.5 | 0.125 | 4 | 0.125 | 4 |
| 214,436 | 0.2 | >16 | 0.25 | 5 | 1 | 0.25 | 0.125 | 4 | 0.125 | 0.125 |
| 214,684 | 0.2 | >16 | 0.25 | 2.5 | 1 | 0.5 | 0.125 | 4 | 0.125 | 0.25 |
| 214,970 | >3.2 | >16 | >32 | 2.5 | 0.5 | 0.25 | 0.125 | 4 | 0.125 | 0.125 |
| 215,008 | 3.2 | >16 | >32 | 2.5 | 0.125 | 0.125 | 0.125 | 4 | 0.125 | 0.25 |
| 215,061 | 3.2 | >16 | >32 | 2.5 | 0.5 | 0.25 | 0.125 | 4 | 0.125 | 0.25 |
| 215,184 | 0.4 | >16 | 0.5 | 0.32 | 0.25 | 0.125 | 0.125 | 4 | 0.125 | 0.25 |
| 215,205 | 0.8 | 0.25 | 4 | 10 | 2 | 0.5 | 0.125 | 2 | 0.125 | 0.125 |
| 215,287 | >3.2 | >16 | 0.25 | 10 | 16 | 4 | 0.25 | 8 | 0.25 | 0.25 |
| 215,338 | 0.8 | >16 | >32 | 2.5 | 4 | 2 | 0.125 | 4 | 0.125 | 0.5 |
| 215,428 | 1.6 | 4 | >32 | 1.25 | 4 | 1 | 0.5 | 8 | 0.5 | 1 |
| 215,655 | 0.2 | >16 | >32 | 5 | 0.25 | 0.125 | 0.25 | 4 | 0.25 | 0.25 |
| 215,682 | 0.8 | >16 | >32 | 1.25 | 0.5 | 0.25 | 0.125 | 4 | 0.125 | 0.25 |
| 215,854 | 0.025 | 0.125 | >32 | 0.63 | 0.25 | 0.125 | 0.125 | 8 | 0.125 | 0.25 |
| 220,153 | >3.2 | >16 | >32 | 5 | 4 | 1 | 0.125 | 4 | 0.125 | 0.25 |
| 220,324 | 1.6 | 0.125 | 1 | 0.63 | >16 | >16 | 0.125 | 4 | 0.125 | 0.5 |
| 220,459 | 0.025 | 0.125 | 0.25 | 0.32 | 4 | 2 | 0.125 | 4 | 0.125 | 0.5 |
| 220,532 | 0.4 | >16 | 32 | 2.5 | 4 | 2 | 0.125 | 4 | 0.125 | 0.25 |
| 220,559 | 0.025 | 0.125 | 0.25 | 0.63 | 4 | 2 | 0.125 | 4 | 0.125 | 0.5 |
| 220,758 | 0.4 | >16 | >32 | 0.63 | 1 | 0.5 | 0.125 | 4 | 0.125 | 0.5 |
| 220,660 | 3.2 | >16 | >32 | 5 | 0.125 | 0.125 | 0.125 | 4 | 0.125 | 0.25 |
| 220,966 | 3.2 | >16 | >32 | 2.5 | 0.25 | 0.125 | 0.125 | 4 | 0.125 | 0.25 |
| 221,209 | 3.2 | >16 | >32 | 10 | 2 | 1 | 0.124 | 4 | 0.124 | 0.5 |

[a]MRX-I, conezolid; INH, isoniazid; RFP, rifampicin; SM, streptomycin; EMB, ethambutol; LFX, levofloxacin; MXF, moxifloxacin; CFZ, clofazimine; CS, cycloserine; LZD, linezolid.

The MICs of MRX-I for the treatment of drug-resistant *M. tuberculosis* ranged from 0.125 to 4 µg/mL. The detailed MIC distribution for all clinical isolates was shown in Table 2. It was found that, for those isolates that were resistant to first-line anti-*M. tuberculosis* drugs, MRX-I showed good antibacterial properties. Additionally, MRX-I also had a better effect on drug-resistant *M. tuberculosis* than the four second-generation drugs LFX, MXF, CFZ, and CS (Table 2).

The results in this study demonstrated that MRX-I was an efficacious anti-*M. tuberculosis* drug with activity against drug-sensitive and drug-resistant isolates *in vitro*. Notably, it had advantages against isolates that were resistant to frequently used *M. tuberculosis* treatment drugs. Its activity was comparable to that of another efficient

oxazolidinone, LZD. Therefore, MRX-I might be a novel and prospective anti-*M. tuberculosis* drug.

## AUTHOR AFFILIATIONS

[1]Senior Department of Tuberculosis, Tuberculosis Prevention and Control Key Laboratory/Beijing Key Laboratory of New Techniques of Tuberculosis Diagnosis and Treatment, The Eighth Medical Center of PLA General Hospital, Beijing, China

[2]Department of Emergency, The Eighth Medical Center of PLA General Hospital, Beijing, China

[3]Department of Research and Training, The Eighth Medical Center of PLA General Hospital, Beijing, China

## AUTHOR ORCIDs

Xiao Liu  http://orcid.org/0009-0004-9085-2166
Jianqin Liang  http://orcid.org/0000-0001-9173-7101

## FUNDING

| Funder | Grant(s) | Author(s) |
| --- | --- | --- |
| Beijing Municipal Administration of Hospitals Clinical Medicine Development of Special Funding Support | Z171100001017187 | Huiru An |

## AUTHOR CONTRIBUTIONS

Huiru An, Conceptualization, Data curation, Funding acquisition, Investigation, Methodology, Writing – original draft | Wenna Sun, Formal analysis, Investigation, Methodology | Xiao Liu, Formal analysis, Investigation, Writing – original draft | Tianhao Wang, Methodology, Validation, Writing – original draft | Juan Qiao, Project administration, Resources, Supervision, Validation, Writing – review and editing | Jianqin Liang, Conceptualization, Supervision, Validation, Writing – review and editing

## ADDITIONAL FILES

The following material is available online.

Open Peer Review

**PEER REVIEW HISTORY (review-history.pdf).** An accounting of the reviewer comments and feedback.

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
