## [Reviewer comments · Microbiology Spectrum]

Microbiology Spectrum

In vitro activities of contezolid (MRX-I) against drug-sensitive and drug-resistant *Mycobacterium tuberculosis*

Huiru An, Wenna Sun, Xiao Liu, Tianhao Wang, Juan Qiao, and Jianqin Liang

Corresponding Author(s): Jianqin Liang, Tuberculosis Prevention and Control Key Laboratory/Beijing Key Laboratory of New Techniques of Tuberculosis Diagnosis and Treatment, Senior Department of Tuberculosis, The Eighth Medical Center of PLA General Hospital

Review Timeline:

Submission Date:	January 17, 2023
Editorial Decision:	April 24, 2023
Revision Received:	July 6, 2023
Accepted:	July 24, 2023

Editor: Rosemary She

Reviewer(s): Disclosure of reviewer identity is with reference to reviewer comments included in decision letter(s). The following individuals involved in review of your submission have agreed to reveal their identity: Fulya BAYINDIR BİLMAN (Reviewer #2)

Transaction Report:

DOI: <https://doi.org/10.1128/spectrum.04627-22>

April 24, 2023

Dr. Jianqin Liang

Tuberculosis Prevention and Control Key Laboratory/Beijing Key Laboratory of New Techniques of Tuberculosis Diagnosis and Treatment, Senior Department of Tuberculosis, The Eighth Medical Center of PLA General Hospital
Beijing 100091
China

Re: Spectrum04627-22 (In vitro activities of contezolid (MRX-I) against drug-sensitive and drug-resistant Mycobacterium tuberculosis)

Dear Dr. Jianqin Liang:

Thank you for submitting your manuscript to Microbiology Spectrum. Your manuscript has been reviewed by 2 experts in the field. Our consensus was that the submission has strong merit, but some revisions are required and hence the decision is Modifications. When submitting the revised version of your paper, please provide (1) point-by-point responses to the issues raised by the reviewers and editor as file type "Response to Reviewers," not in your cover letter, and (2) a PDF file that indicates the changes from the original submission (by highlighting or underlining the changes) as file type "Marked Up Manuscript - For Review Only". Please use this link to submit your revised manuscript - we strongly recommend that you submit your paper within the next 60 days or reach out to me. Detailed instructions on submitting your revised paper are below.

Link Not Available

Sincerely,

Rosemary She

Journals Department
Reviewer comments:

Reviewer #1 (Comments for the Author):

1-More detailed information should be given about the Roche drug susceptibility testing used in the study.
2-In Table 5, LZD MICs are observed to be lower than MRX-I MICs. Therefore, the advantages of MRX-I over LZD should be mentioned. It is also known whether there is cross-resistance between the two antibiotics.

Reviewer #2 (Comments for the Author): no comments provided

Editor comments:

-Abstract, the susceptibility testing is described as the Roche method but should mention the actual method, e.g. broth

microdilution, agar proportion, etc. This could be in addition to or instead of the commercial name.

-Agree with Reviewer #1, please define the Roche method with enough clarity that the method can be replicated by someone else. Please also clarify line 86, method of MIC assays. Is this the same or different from the Roche method- this is not clear. If MIC method is different, provide enough details/reference such that the reader could replicate the method used.

-Line 55, do you mean the drug was developed "by China" as in the country/government, or "in China" to be more general? Please also provide some more information about contezolid if for already approved indications, MIC breakpoints in China are comparable to linezolid.

-Line 68, please ensure statement of IRB approval or waiver (not necessarily informed consent) is included in the submission according to ASM journal policies.

-Line 69, add reference for CLSI guideline as to indicate which guideline was followed.

-Line 78, clarify/define "drug-resistant." It seems to mean resistance to one or more of the agents tested; regardless define briefly in the text.

-Line 84, clarify why there were clinical isolates from 50 patients to start (line 66) but only 48 with Roche susceptibility results.

-Line 85, clarify what "drug sensitive" means, perhaps "pan-drug sensitive," if appropriate, can be stated in this line.

-Line 90, suggest "higher" instead of "larger" for MIC comparisons.

-Lines 91-93, clarify meaning of "which are replaceable" otherwise remove this phrase if appropriate.

-Lines 109-110, define "as effective" given that no statistical analysis was performed (see Reviewer #1 comments) and no clinical data are available. Similar, Line 111, clarify what the intended meaning is of "efficient" when referring to MIC studies. Do the authors mean "efficacious" or similar?

-Consider adding MIC50 and MIC90 analysis for linezolid and contezolid to text and tables.

-Please condense the Tables to reduce redundancy. Table 1, the grading of 1-4+ should be explained, and Table 1 should be moved to supplementary materials. Table 2, caption appears to be incorrect and identical to Table 3 caption. Regardless, either combined Tables 2 and 3 together, and Tables 4 and 5 together; or, combine Tables 2 and 5 together, and Tables 3 and 4 together.

Staff Comments:

Preparing Revision Guidelines

Please return the manuscript within 60 days; if you cannot complete the modification within this time period, please contact me. If you do not wish to modify the manuscript and prefer to submit it to another journal, please notify me of your decision immediately so that the manuscript may be formally withdrawn from consideration by Microbiology Spectrum.

**Title:** In vitro activities of contezolid (MRX-I) against drug-sensitive and drug-resistant
*Mycobacterium tuberculosis*

**Authors:** Huiru An¹†, MD, Wenna Sun¹†, MD, Xiao Liu¹†, MD, Tianhao Wang^{*2}, MD,
Juan Qiao^{*3}, MD, Jianqin Liang^{*1}

**Affiliations:**

1 Tuberculosis Prevention and Control Key Laboratory/Beijing Key Laboratory of New

Techniques of Tuberculosis Diagnosis and Treatment, Senior Department of Tuberculosis,

The Eighth Medical Center of PLA General Hospital, Beijing 100091, China

2 Department of Emergency, The Eighth Medical Center of PLA General Hospital, Beijing

100091, China

3 Department of Research and Training, The Eighth Medical Center of PLA General Hospital,

Beijing 100091, China.

† These authors contributed equally to this work. Author order was determined by drawing

straws.

*** Correspondence:**

Tianhao Wang¹, wt19750728@hotmail.com; Juan Qiao², qiaojuan775008@sina.com; and

Jianqin Liang³, ljbj309@163.com.

**Abstract**

A novel oxazolidinone for the treatment of *Mycobacterium tuberculosis* has been
developed, but the activity of contezolid (MRX-I) still needs to be clarified. In this
study, we isolated *Mycobacterium tuberculosis* from 48 clinical patients with

pulmonary tuberculosis. Roche drug susceptibility tests identified 9 drug-sensitive
and 39 drug-resistant *Mycobacterium tuberculosis* isolates. Drug susceptibility assays
indicated that MRX-I exhibited anti-*Mycobacterium tuberculosis* activity against both
drug-sensitive and drug-resistant isolates, with an advantage against drug-resistant
isolates. The results also showed that the anti-*Mycobacterium tuberculosis* activity
was comparable to that of linezolid (LZD).

**Keywords** *Mycobacterium tuberculosis*, Contezolid, in vitro, drug-resistant

**Importance**

Currently, *Mycobacterium tuberculosis* (MTB) has exhibited increased drug
resistance, leading to ineffective drug treatment in many patients with tuberculosis.
Among the anti-*Mycobacterium tuberculosis* drugs, oxazolidinones have been
gradually developed. Contezolid (MRX-I) has been newly developed in China with
advantages versus the first oxazolidinone antibiotic approved by the Food and Drug
Administration (FDA) for clinical use, but the anti-*Mycobacterium tuberculosis*
activity needs to be further clarified. In this study, in vitro activities of MRX-I against
MTB were tested. The drug susceptibility assays indicated that MRX-I exhibited
anti-*Mycobacterium tuberculosis* activity comparable to that of linezolid (LZD), with
an advantage against drug-resistant isolates.

The widespread use of various antibiotics has inevitably led to the increasingly
serious problem of bacterial drug resistance, and the emergence of multidrug-resistant
(MDR) bacteria poses a major challenge to public health systems in all countries
worldwide. *Mycobacterium tuberculosis* (MTB), the pathogenic agent of pulmonary
tuberculosis (TB), has exhibited increased drug resistance, leading to ineffective drug
treatment in many patients with tuberculosis (1-2). Among the anti-*Mycobacterium*
*tuberculosis* drugs, oxazolidinones, a novel synthetic antibacterial drug class, have
been gradually developed (3-5). Among the oxazolidinones, linezolid (LZD) has a
strong antibacterial effect and was the first oxazolidinone antibiotic approved by the
Food and Drug Administration (FDA) for clinical use (6-8). However, LZD use can
cause adverse reactions, such as anti-tuberculosis drug-induced liver injury (ADLI),
bone marrow suppression and peripheral neuropathy (9).

Contezolid (MRX-I), a new oxazolidinone independently developed by China, has
been approved for distribution on the clinical market. MRX-I has a strong
anti-gram-positive bacterial ability and has an efficacy similar to that of LZD (10-11).
However, compared with LZD, MRX-I has more safety advantages, including
tolerance and minimal adverse effects (such as hematological adverse events and
myelosuppression) (12). Additionally, it has been reported that MRX-I exhibits
anti-*Mycobacterium tuberculosis* activity in vitro/vivo and rarely induces drug
resistance, so it could be a prospective drug for *Mycobacterium tuberculosis* infection
treatment (13). However, for drug-resistant *Mycobacterium tuberculosis*, the efficacy
of and susceptibility to MRX-I remain to be clarified.

**Isolation and validation of drug-resistant *Mycobacterium tuberculosis***

Clinical *Mycobacterium tuberculosis* isolates were collected from 50 different
patients with pulmonary tuberculosis. The specimen types included lung lavage fluid
and sputum. All patients signed informed consent forms. The procedure followed the
Clinical and Laboratory Standards Institute guidelines. Subsequently, drug

susceptibility testing using Roche *Mycobacterium tuberculosis* medium was
performed for all isolates to analyze drug resistance status in the *Mycobacterium*
*tuberculosis* isolates.

The drug susceptibility tests included 4 common first-line anti-*Mycobacterium*
*tuberculosis* drugs, isoniazid (INH), streptomycin (SM), ethambutol (EMB), and
rifampicin (RFP), and 8 other second-generation anti-*Mycobacterium tuberculosis*
drugs, including levofloxacin (LFX), kanamycin (KAN), capreomycin (CPM),
prothionamide (TH1321), p-aminosalicylic acid (PAS), rifapentine (RFT), rifabutin
(RBU), and amikacin (AMK). It was found that of the 39 drug-resistant
*Mycobacterium tuberculosis* isolates, 82.0% (32/39) exhibited drug resistance to
first-line anti-*Mycobacterium tuberculosis* drugs, and 100.0% (39/39) exhibited drug
resistance to the other second-generation drugs (**Table 1**).

**Contezolid exhibited anti-*Mycobacterium tuberculosis* activity against**
**drug-sensitive *Mycobacterium tuberculosis*.**

According to the Roche drug susceptibility test results of 48 clinical isolates, 9
isolates were drug sensitive. These drug-sensitive *Mycobacterium tuberculosis*
isolates were then subjected to minimum inhibitory concentration (MIC) assays. The
MIC of MRX-I ranged from 0.25-0.5 µg/ml. In addition, compared with another
oxazolidinone, LZD, MRX-I exhibited anti-*Mycobacterium tuberculosis* activity that
was comparable to that of LZD (**Table 2**).

The MRX-I MIC value was larger than those of the other drugs, except CS and EMB,
for the drug-sensitive *Mycobacterium tuberculosis* isolates. This result indicates that
MRX-I has no significant advantage in the treatment of drug-sensitive
*Mycobacterium tuberculosis*, which are replaceable (**Table 3**).

**MRX-I exhibits anti-*Mycobacterium tuberculosis* activity against drug-resistant**
***Mycobacterium tuberculosis*.**

The MICs of MRX-I for the treatment of drug-resistant *Mycobacterium tuberculosis*
ranged from 0.125 to 4 µg/ml. The detailed MIC distribution for all clinical isolates is
shown in Table 4. It was found that, for those isolates that were resistant to first-line
anti-*Mycobacterium tuberculosis* drugs, MRX-I showed good antibacterial properties.
Additionally, MRX-I also had a better effect on drug-resistant *Mycobacterium*
*tuberculosis* than the four second-generation drugs LFX, MXF, CFZ and CS (Table
4).

LZD is another oxazolidinone that has strong activity against *Mycobacterium*
*tuberculosis*. To date, it has been widely used in the treatment of drug-resistant
tuberculosis. There is evidence suggesting that adding it to the treatment regimen
could improve the negative conversion rate and cure rate. However, the adverse
effects caused by LZD are difficult to address. Therefore, alternative drugs are
urgently needed. The MICs of MRX-I compared with those of LZD against
drug-resistant *Mycobacterium tuberculosis* are presented in Table 5. MRX-I was as
effective as LZD against these drug-resistant *Mycobacterium tuberculosis* isolates.

These results demonstrated that MRX-I was an efficient anti-*Mycobacterium*
*tuberculosis* drug with activity against drug-sensitive and drug-resistant isolates in
vitro. Notably, it had advantages against isolates that were resistant to frequently used
*Mycobacterium tuberculosis* treatment drugs. Its activity was comparable to that of
another efficient oxazolidinone, LZD. Therefore, MRX-I might be a novel and
prospective anti-*Mycobacterium tuberculosis* drug.

REFERENCES

- 1. Ma L, Gao M. 2022. Analysis of clinical characteristics and risk factors for drug
resistance in newly-treated patients with pulmonary tuberculosis complicated with
chronic obstructive pulmonary disease. *Infect Drug Resist* 15:4861-4869.
- 2. Merker M, Rasigade JP, Barbier M, Cox H, Feuerriegel S, Kohl TA, Shitikov E,
Klaos K, Gaudin C, Antoine R, Diel R, Borrell S, Gagneux S, Nikolayevskyy V,

- Andres S, Crudu V, Supply P, Niemann S, Wirth T. 2022. Transcontinental spread
and evolution of *Mycobacterium tuberculosis* W148 European/Russian clade
toward extensively drug resistant tuberculosis. *Nat Commun* 13(1):5105.
- 3. Conradie F, Diacon AH, Ngubane N, Howell P, Everitt D, Crook AM, Mendel
CM, Egizi E, Moreira J, Timm J, McHugh TD, Wills GH, Bateson A, Hunt R,
Van Niekerk C, Li M, Olugbosi M, Spigelman M. 2020. Treatment of Highly
Drug-Resistant Pulmonary Tuberculosis. *N Engl J Med* 382(10):893-902.
- 4. Kaushik A, Heuer AM, Bell DT, Culhane JC, Ebner DC, Parrish N, Ippoliti JT,
Lamichhane G. 2016. An evolved oxazolidinone with selective potency against
*Mycobacterium tuberculosis* and gram positive bacteria. *Bioorg Med Chem Lett*
26(15):3572-3576.
- 5. Zhao H, Wang B, Fu L, Li G, Lu H, Liu Y, Sheng L, Li Y, Zhang B, Lu Y, Ma C,
Huang H, Zhang D, Lu Y. 2020. Discovery of a Conformationally Constrained
Oxazolidinone with Second-generation Safety and Efficacy Profiles for the
Treatment of Multidrug-Resistant Tuberculosis. *J Med Chem* 63(17):9316-9339.
- 6. Thwaites G, Nguyen NV. 2022. Linezolid for Drug-Resistant Tuberculosis. *N*
*Engl J Med* 387(9):842-843.
- 7. Zhao W, Zheng M, Wang B, Mu X, Li P, Fu L, Liu S, Guo Z. 2016. Interactions
of linezolid and second-line anti-tuberculosis agents against multidrug-resistant
*Mycobacterium tuberculosis* in vitro and in vivo. *Int J Infect Dis* 52:23-28.
- 8. Maltempe FG, Caleffi-Ferracioli KR, do Amaral RCR, de Oliveira Demitto F,
Siqueira VLD, de Lima Scodro RB, Hirata MH, Pavan FR, Cardoso RF. 2017.
Activity of rifampicin and linezolid combination in *Mycobacterium*
*tuberculosis*. *Tuberculosis (Edinb)* 104:24-29.
- 9. Kishor K, Dhasmana N, Kamble SS, Sahu RK. 2015. Linezolid Induced Adverse
Drug Reactions - An Update. *Curr Drug Metab* 16(7):553-559.
- 10. Hoy SM. 2021. Correction to: Contezolid: First Approval. *Drugs* 81(16):1945.

- 11. Carvalhaes CG, Duncan LR, Wang W, Sader HS. 2020. In Vitro Activity and
Potency of the Novel Oxazolidinone Contezolid (MRX-I) Tested against
Gram-Positive Clinical Isolates from the United States and Europe. *Antimicrob*
*Agents Chemother* 64(11):e01195-20.
- 12. Yang M, Zhan S, Fu L, Wang Y, Zhang P, Deng G. 2022. Prospects of contezolid
(MRX-I) against multidrug-resistant tuberculosis and extensively drug-resistant
tuberculosis. *Drug Discov Ther* 16(2):99-101.
- 13. Guo Q, Xu L, Tan F, Zhang Y, Fan J, Wang X, Zhang Z, Li B, Chu H. 2021.
Novel Oxazolidinone, Contezolid (MRX-I), Expresses Anti-Mycobacterium
abscessus Activity In Vitro. *Antimicrob Agents Chemother* 65(11):e0088921.

**TABLE 1** Results of 39 drug-resistant *Mycobacterium tuberculosis* isolates subjected to Roche drug
 susceptibility testing ^a.

	INH		SM		EMB		RFP		LFX		KAN		CPM		TH1321		PAS		RFT		RBU		AMK			
	L	H	L	H	L	H	L	H	L	H	L	H	L	H	L	H	L	H	L	H	L	H	L	H	L	H
191160												1+						1+							1+	
191163	4+																	1+								
191186	4+		4+	4+			4+	4+				1+						3+	1+	4+	4+	4+	3+			
191191																		1+							1+	
191209																		1+								
191247																		1+		1+						
191269			4+	4+						4+	1+															
191396	4+		4+	3+			4+	4+												2+	2+	1+				
191524	3+		3+	1+			3+	1+												3+	3+	2+	1+			
191554										4+								1+								
201466	4+	4+	4+	4+	4+		4+	4+	4+	1+	4+	4+								4+	4+			4+		
213829	4+		4+	1+						4+								1+	1+							
214014	4+		4+	4+	4+		2+		2+											2+						
214022	2+		4+	4+	1+		4+	4+	4+		1+	1+						1+		4+	4+	4+	2+	1+		
214124	4+		4+	4+			4+	4+	4+											4+	4+	4+	1+			
214225			4+	4+	1+		1+		4+	1+																
214320					2+		1+		4+	1+										4+	4+	4+				
214436					1+		1+	1+	1+											1+	1+	1+				
214684					1+		1+	1+	1+											1+	1+	1+				
214970	2+		2+	2+			1+	1+			1+	1+						1+	1+	2+	2+			2+	2+	
215008	1+		1+	1+			1+	1+																		
215061	2+		2+	2+			1+	1+			1+	1+						1+	1+	2+	2+			2+	2+	
215184							2+	2+												2+	2+	1+	1+			
215205	1+		2+		1+				4+	1+								4+	2+	1+						
215287	4+				1+		4+	4+	4+	1+										4+	4+	4+	4+			
215338	1+		4+	2+			4+	4+	4+	3+								2+		4+	4+	4+	4+			
215428	4+		4+	4+			4+	4+	4+	3+	4+	4+	4+	3+				3+	2+	4+	4+	3+		4+	4+	
215655			1+	1+			1+	1+												1+	1+	1+	1+			
215682	1+						1+	1+	1+											1+	1+	1+	1+			
215854			2+	2+																						
220153	4+		4+	4+	1+		4+	4+	2+	1+										4+	4+	1+				
220324	4+									4+	4+															
220459										4+	3+							1+								
220532	4+		4+	3+			4+	4+	4+	4+											4+	4+	4+	4+		
220559										4+	1+															
220758			4+	4+			4+	4+	4+												4+	4+	4+	2+		
220660	4+		4+	4+	2+		4+	4+			4+	4+			1+						4+	4+	4+		4+	4+
220966	4+		4+	4+	2+		4+	4+													4+	4+	4+	4+		

*a* INH, isoniazid; SM, streptomycin; EMB, ethambutol; RFP, rifampicin; LFX, levofloxacin; KAN,
 kanamycin; CPM, capreomycin; TH1321, prothionamide; PAS, p-aminosalicylic acid; RFT,
 rifapentine; RBU, rifabutin; AMK, amikacin; L, low concentration; H, concentration.

**TABLE 2** MICs of MRX-I compared to those of INH, RFP, SM, EMB, LFX, moxifloxacin (MXF),
 clofazimine (CFZ) and cycloserine (CS) against 9 isolates of drug-sensitive *Mycobacterium*
 *tuberculosis*^a.

Mycobacterium tuberculosis isolate	MIC (µg/ml)	
	LZD	MRX-I
215680	0.25	0.5
220213	0.25	0.25
220220	0.25	0.5
220311	0.5	0.5
220391	0.25	0.25
220412	0.25	0.25
220450	0.5	0.5
220509	0.5	0.5
220667	0.25	0.5

a MRX-I, conezolid; LZD, linezolid.

**TABLE 3** MICs of MRX-I compared to those of INH, RFP, SM, EMB, LFX, MXF, CFZ and CS

against 9 isolates of drug-sensitive *Mycobacterium tuberculosis*^a.

Mycobacterium tuberculosis isolate	MIC (µg/ml)								
	INH	RFP	SM	EMB	LFX	MXF	CFZ	CS	MRX-I
215680	0.025	0.25	0.5	0.625	0.25	0.125	0.125	4	0.5
220213	0.025	0.125	0.25	0.625	0.25	0.125	0.125	4	0.25
220220	0.025	0.125	0.25	0.625	0.25	0.125	0.125	8	0.5
220311	0.025	0.125	0.25	0.625	0.25	0.125	0.25	8	0.5
220391	0.025	0.125	0.25	0.625	0.25	0.125	0.25	4	0.25
220412	0.025	0.125	0.25	0.625	0.25	0.125	0.125	4	0.25
220450	0.025	0.125	0.25	0.625	0.25	0.125	0.125	8	0.5
220509	0.025	0.25	0.25	0.625	0.25	0.125	0.125	2	0.5
220667	0.2	1	0.5	0.625	0.25	0.125	0.25	8	0.5

175 ^a MRX-I, conezolid; INH, isoniazid; RFP, rifampicin; SM, streptomycin; EMB, ethambutol; LFX,
 levofloxacin; MXF, moxifloxacin; CFZ, clofazimine; CS, cycloserine.
 **TABLE 4** MICs of MRX-I compared to those of INH, RFP, SM, EMB, LFX, MXF, CFZ and CS
 against 39 isolates of drug-resistant *Mycobacterium tuberculosis*^a.

Mycobacterium tuberculosis isolate	MIC (µg/ml)								
	INH	RFP	SM	EMB	LFX	MXF	CFZ	CS	MRX-I
191160	0.05	0.25	0.5	1.25	0.25	0.125	0.125	4	0.5
191163	1.6	0.5	0.5	0.63	0.25	0.125	0.125	4	0.5
191186	0.8	>16	>32	0.63	0.125	0.125	0.125	2	0.25
191191	0.025	0.125	0.25	0.63	0.25	0.125	0.125	4	0.25
191209	0.025	0.25	0.25	0.63	0.25	0.125	0.125	4	0.25
191247	1.6	0.5	0.5	2.5	0.125	0.125	0.125	4	0.5
191269	0.025	0.125	>32	0.63	4	2	0.125	4	0.25
191396	1.6	>16	>32	0.63	0.125	0.125	0.125	4	0.25
191524	3.2	>16	32	2.5	0.25	0.125	0.125	4	0.25
191554	0.025	0.125	0.25	0.63	0.5	0.25	0.125	4	0.25
201466	>3.2	2	>32	5	2	1	0.5	4	0.125
213829	3.2	0.125	8	0.63	2	1	0.125	4	0.5
214014	3.2	0.25	>32	10	1	0.125	0.5	4	0.125
214022	0.8	8	2	5	2	1	0.125	2	0.125
214124	0.8	>16	>32	5	4	2	0.125	4	0.125
214225	0.1	16	>32	5	2	0.5	0.125	4	0.5
214320	0.025	>16	0.25	5	2	0.5	0.125	4	4
214436	0.2	>16	0.25	5	1	0.25	0.125	4	0.125
214684	0.2	>16	0.25	2.5	1	0.5	0.125	4	0.25
214970	>3.2	>16	>32	2.5	0.5	0.25	0.125	4	0.125
215008	3.2	>16	>32	2.5	0.125	0.125	0.125	4	0.25
215061	3.2	>16	>32	2.5	0.5	0.25	0.125	4	0.25
215184	0.4	>16	0.5	0.32	0.25	0.125	0.125	4	0.25
215205	0.8	0.25	4	10	2	0.5	0.125	2	0.125
215287	>3.2	>16	0.25	10	16	4	0.25	8	0.25
215338	0.8	>16	>32	2.5	4	2	0.125	4	0.5

215428	1.6	4	>32	1.25	4	1	0.5	8	1
215655	0.2	>16	>32	5	0.25	0.125	0.25	4	0.25
215682	0.8	>16	>32	1.25	0.5	0.25	0.125	4	0.25
215854	0.025	0.125	>32	0.63	0.25	0.125	0.125	8	0.25
220153	>3.2	>16	>32	5	4	1	0.125	4	0.25
220324	1.6	0.125	1	0.63	>16	>16	0.125	4	0.5
220459	0.025	0.125	0.25	0.32	4	2	0.125	4	0.5
220532	0.4	>16	32	2.5	4	2	0.125	4	0.25
220559	0.025	0.125	0.25	0.63	4	2	0.125	4	0.5
220758	0.4	>16	>32	0.63	1	0.5	0.125	4	0.5
220660	3.2	>16	>32	5	0.125	0.125	0.125	4	0.25
220966	3.2	>16	>32	2.5	0.25	0.125	0.125	4	0.25
221209	3.2	>16	>32	10	2	1	0.124	4	0.5

180 ^a MRX-I, conezolid; INH, isoniazid; RFP, rifampicin; SM, streptomycin; EMB, ethambutol; LFX,
levofloxacin; MXF, moxifloxacin; CFZ, clofazimine; CS, cycloserine.

**TABLE 5** MICs of MRX-I compared to those of the classical oxazolidinone LZD against 39

isolates of drug-resistant *Mycobacterium tuberculosis*.

Mycobacterium tuberculosis isolate	MIC (µg/ml)	
	LZD	MRX-I
191160	0.125	0.5
191163	0.125	0.5
191186	0.125	0.25
191191	0.125	0.25
191209	0.125	0.25
191247	0.125	0.5
191269	0.125	0.25
191396	0.125	0.25
191524	0.125	0.25
191554	0.125	0.25
201466	0.5	0.125
213829	0.125	0.5
214014	0.5	0.125
214022	0.125	0.125
214124	0.125	0.125
214225	0.125	0.5
214320	0.125	4

214436	0.125	0.125
214684	0.125	0.25
214970	0.125	0.125
215008	0.125	0.25
215061	0.125	0.25
215184	0.125	0.25
215205	0.125	0.125
215287	0.25	0.25
215338	0.125	0.5
215428	0.5	1
215655	0.25	0.25
215682	0.125	0.25
215854	0.125	0.25
220153	0.125	0.25
220324	0.125	0.5
220459	0.125	0.5
220532	0.125	0.25
220559	0.125	0.5
220758	0.125	0.5
220660	0.125	0.25
220966	0.125	0.25
221209	0.124	0.5

185

^a MRX-I, contezolid; LZD, linezolid.

Reviewer comments:

Reviewer #1:

1-More detailed information should be given about the Roche drug susceptibility testing used in the study.

Response: Thank you. More detailed information about the Roche drug susceptibility testing including methods and reagents was provided and added in the text (Lines 76-98, Page 4 and 5).

2-In Table 5, LZD MICs are observed to be lower than MRX-I MICs. Therefore, the advantages of MRX-I over LZD should be mentioned. It is also known whether there is cross-resistance between the two antibiotics.

Response: We agreed with your concern and added related contents to the manuscript as shown in the lines 64-69, page 3 as follows: "Besides, compared with LZD, MRX-I has more safety advantages, including tolerance and minimal adverse effects (such as hematological adverse events and myelosuppression) (12). Additionally, it has been reported that MRX-I exhibits anti-Mycobacterium tuberculosis activity in vitro/vivo and rarely induces drug resistance, so it could be a prospective drug for Mycobacterium tuberculosis infection treatment (13)". Additional solid evidence would be provided to prove the cross-resistance and necessitate additional research.

Reviewer #2 :

Editor comments:

-Abstract, the susceptibility testing is described as the Roche method but should mention the actual method, e.g. broth microdilution, agar proportion, etc. This could be in addition to or instead of the commercial name.

Response: Thank you for your advice. We have added the detailed information into the manuscript(Lines 76-98, Page 4 and 5).

-Agree with Reviewer #1, please define the Roche method with enough clarity that the method can be replicated by someone else. Please also clarify line 86, method of MIC assays. Is this the same or different from the Roche method- this is not clear. If MIC method is different, provide enough details/reference such that the reader could replicate the method used.

Response: Thank you. We have added more information to define and clarify the Roche method and MIC assays as shown in the Materials and Methods paragraph as shown from line 76-98, Page 4 and 5.

-Line 55, do you mean the drug was developed "by China" as in the country/government, or "in China" to be more general? Please also provide some more information about contezolid if for already approved indications, MIC breakpoints in China are comparable to linezolid.

Response: Thank you for your suggestion. We have changed the words "by China" into "in China". We also added more information about contezolid as shown in line 56-64, page 3, as follows, "According to National Medical Products Administration (NMPA), MRX-I was recommended for the treatment of acute skin and soft tissue infections caused by gram-positive bacteria, including methicillin resistant *Staphylococcus aureus* (MRSA) and vancomycin resistant *Enterococcus* (VRE). MRX-I had a strong anti-gram-positive bacterial activity with a similar efficacy to that of LZD, and the clinical cure rates of MRX-I and LZD were reported to be 93.0% and 93.4% at post treatment visits respectively in a critical phase III study (CTR20150855) for the treatment of complex skin and soft tissue infections in China which was conducted in 50 clinical centers (10-11)."

-Line 68, please ensure statement of IRB approval or waiver (not necessarily informed consent) is included in the submission according to ASM journal policies.

Response: Thank you. We have added the statement of IRB into the method part as shown in lines 73-74, page 4.

-Line 69, add reference for CLSI guideline as to indicate which guideline was followed.

Response: Thank you. The guideline was briefly explained with a reference cited (Lines 76-80, Page 4).

-Line 78, clarify/define "drug-resistant." It seems to mean resistance to one or more of the agents tested; regardless define briefly in the text.

Response: Thank you. We have added brief definition "which were resistant to one or more of the agents tested" into the sentence to make the meaning more clear, as shown in the lines 100-101, page 5.

-Line 84, clarify why there were clinical isolates from 50 patients to start (line 66) but only 48 with Roche susceptibility results.

Response: We apologize for the error and have corrected the mistake in line 104, page 5.

-Line 85, clarify what "drug sensitive" means, perhaps "pan-drug sensitive," if appropriate, can be stated in this line.

Response: We agree with your comment. Here, "drug sensitive" means "pan-drug sensitive". We have made the revision shown in the line 105, page 5.

-Line 90, suggest "higher" instead of "larger" for MIC comparisons.

Response: Thank you. We have changed the word "larger" into "higher" as shown in the line 108, page 5.

-Lines 91-93, clarify meaning of "which are replaceable" otherwise remove this phrase if appropriate.

Response: Thank you for your advice. We have removed this phrase.

-Lines 109-110, define "as effective" given that no statistical analysis was performed (see Reviewer #1 comments) and no clinical data are available. Similar, Line 111, clarify what the intended meaning is of "efficient" when referring to MIC studies. Do the authors mean "efficacious" or similar?

Response: Thank you. We accepted your suggestion and made corresponding revisions. The improper expressions "MRX-I was as efficacious as LZD against these drug-resistant Mycobacterium tuberculosis isolates" were removed, and the word "effective" was changed to "efficacious".

-Consider adding MIC50 and MIC90 analysis for linezolid and contezolid to text and tables.

Response: Thank you. Analysis for MIC50 and MIC90 has been added as follows (lines 89-91, page 4): "In the data of this study, the MICs values we showed were MIC90. Linezolid and Conteozolid showed consistent data for MIC50 and MIC90 of Mycobacterium tuberculosis isolates, with values of 0.25ug/ml and 0.5ug/ml, respectively".

-Please condense the Tables to reduce redundancy. Table 1, the grading of 1-4+ should be explained, and Table 1 should be moved to supplementary materials. Table 2, caption appears to be incorrect and identical to Table 3 caption. Regardless, either combined Tables 2 and 3 together, and Tables 4 and 5 together; or, combine Tables 2 and 5 together, and Tables 3 and 4 together.

Response: Thank you. We have rearranged all the Tables according to your advice. The original Table 1 has been removed from the manuscript and added to supplementary materials as supplementary Table 1. The grading of 1-4+ has also been illustrated in supplementary Table 2. In addition, original Tables 2 and 3 has been combined as new Table 1, and the original Tables 4 and 5 has been combined as new Table 2.

July 24, 2023

Dr. Jianqin Liang

Tuberculosis Prevention and Control Key Laboratory/Beijing Key Laboratory of New Techniques of Tuberculosis Diagnosis and Treatment, Senior Department of Tuberculosis, The Eighth Medical Center of PLA General Hospital
Beijing 100091
China

Re: Spectrum04627-22R1 (In vitro activities of contezolid (MRX-I) against drug-sensitive and drug-resistant Mycobacterium tuberculosis)

Dear Dr. Jianqin Liang:

Your manuscript has been accepted, and I am forwarding it to the ASM Journals Department for publication. You will be notified when your proofs are ready to be viewed.

Sincerely,

Rosemary She
Editor, Microbiology Spectrum
